# Symptoms in Health Care Workers during the COVID-19 Epidemic. A Cross-Sectional Survey

**DOI:** 10.3390/ijerph17145218

**Published:** 2020-07-20

**Authors:** Nicola Magnavita, Giovanni Tripepi, Reparata Rosa Di Prinzio

**Affiliations:** 1Postgraduate School of Occupational Health, Università Cattolica del Sacro Cuore, 00168 Roma, Italy; repdip@gmail.com; 2Department of Woman/Child & Public Health, Fondazione Policlinico A. Gemelli, 00168 Roma, Italy; 3Research Unit of Reggio Calabria, Institute of Clinical Physiology, Italian National Research Council, 89124 Calabria, Italy; gtripepi@ifc.cnr.it

**Keywords:** occupational epidemiology, anosmia, dysgeusia, occupational stress, occupational disease, organizational justice, effort/reward imbalance, anxiety, depression, sleep quality

## Abstract

In March–April 2020, the Corona Virus Disease 19 (COVID-19) pandemic suddenly hit Italian healthcare facilities and in some of them many staff members became infected. In this work 595 health care workers from a public company were tested for Severe acute respiratory syndrome coronavirus 2 (82 positive) and asked to complete a questionnaire on early COVID-19 symptoms. Respiratory symptoms were present in 56.1% of cases. Anosmia and dysgeusia in COVID-19 cases were found to have an odds ratio (OR) = 100.7 (95% Confidence Interval [CI] = 26.5–382.6) and an OR = 51.8 (95%CI 16.6–161.9), respectively. About one in three of the cases (29.3%) never manifested symptoms. Anxiety was reported by 16.6% of COVID-19 cases and depression by 20.3%, with a significant increase in the estimated risk (OR = 4.3; 95%CI = 2.4–7.4 for anxiety, OR = 3.5; 95%CI = 2.0–6.0 for depression). In cases, sleep was a significant moderating factor in the relationship between occupational stress, or organizational justice, and anxiety. The early diagnosis of COVID-19 in health care workers, must consider, in addition to respiratory disorders and fever, anosmia, dysgeusia, exhaustion, myalgias and enteric disorders. The frequency of anxiety and depression disorders in the population examined was not higher than that commonly recorded in the same company during periodic checks in the years preceding the epidemic. In COVID-19 cases there was a significant risk of anxiety, especially in those who had low sleep quality. Mental health support and improvement interventions must mainly concern workers with positive tests and should also tend to improve sleep quality.

## 1. Introduction

The Corona Virus Disease 19 (COVID-19) pandemic has had a dramatic impact on health care workers (HCWs) all over the world. The need to tackle the new virus as quickly and effectively as possible has led to an unprecedented effort to share scientific information about the disease. Countless studies have been brought to the attention of scientists, researchers and the general public [1,2,3,4,5,6,7,8,9,10].

Although these studies mainly concern patients affected by the disease [11], some describe the situation of the general population [12]. Surprisingly, few health studies have focused on HCWs. Existing studies refer either to the disease induced by Severe acute respiratory syndrome coronavirus 2 (SARS-Cov-2) in HCWs [13] or to psychological effects of previous viral epidemics [14] or current one [15,16], but there are no field studies describing the psychophysical state of HCWs during the acute phase of the epidemic.

One of the vital tasks of the occupational physician who is responsible for preventing diseases and improving the health level of workers is that of promptly detecting early signs of illness and distinguishing them from other important symptoms possibly related to fatigue and stress, but not caused by infection.

In this study we investigated the workers of a Local Health Authority in Central Italy during the sudden outbreak of the COVID-19 epidemic in that region. The abrupt increase in COVID-19 patients forced the hospitals to become a specialist facility for the disease. Within a matter of days, a large number of HCWs came into close unprotected contact with cases of COVID-19 and many of them tested positive for molecular tests performed on nasopharyngeal swab by reverse transcription polymerase chain reaction (RT-PCR).

HCWs are occupationally exposed to biological [17,18] and psychosocial risk factors [19,20,21] that can endanger their health. It is important to promptly identify the early symptoms of infectious diseases and recognize the psychological effects of trauma. The principal aim of this investigation was to evaluate the prevalence of COVID-19 symptoms in a working population. A second aim was to identify the symptoms associated with psychosocial factors in order to consider ways of preventing them.

## 2. Materials and Methods

### 2.1. Population

The population studied was composed of a Local Health Authority that included two hospitals in which, for over 20 years, workers have been monitored for exposure to occupational risks by one of the authors of this study. Over the years, the health surveillance service has systematically carried out health promotion interventions in that workplace using questionnaires designed to measure levels of occupational stress, anxiety, and depression, and to detect sleep problems. Health surveillance was not interrupted during the epidemic, but because of the need to reduce physical contact, information was requested online.

In accordance with the regulations of the Latium Region, all cases of HCWs who had had close unprotected contact with COVID-19 patients were recorded in a Register and, on the basis of the type of contact, underwent RT-PCR on nasopharyngeal swab.

This study was conducted by the physician in charge of health surveillance in the company during the explosion phase of the epidemic and adopted a cross-sectional design to know the health status of HCWs. The focus of the investigation were workers who had unprotected contacts with proven COVID-19 cases. All workers who tested positive for the RT-PCR test (*n* = 90) were contacted. Their health was compared to that of workers who were listed in the aforementioned Register for unprotected exposure, but were negative on the test, and workers who had no unprotected exposure, who were also negative on the test. The workers contacted (810 people) represented 66.3% of the population undergoing health surveillance for occupational risks in the company. Taking into account the fact that 90 HCWs tested positive, we randomly contacted a 2N sample of workers who had been exposed to COVID-19 patients without effective protection but had tested negative at the naso-pharingeal swab, and a 6N sample of workers who had no unprotected exposure. Although selection bias could be never definitively ruled out, controls were comparable, according to the epidemiological principles indicated by Wacholder et al. [22]: study base, as all comparisons were made within the study base; confounding, as the comparisons were not distorted by other factors; accuracy, as all HCWs performed the same test. The workers were contacted by email on 27 March 2020 and invited to provide their data anonymously by means of a questionnaire placed on SurveyMonkey©. On 15 April, a reminder was sent by mail. No incentive was provided for the reply. Data collection was completed on 30 April 2020. 723 people took part in the survey (response rate 89.3%), but only 595 of them, 175 male (29.9%), and 417 female HCWs, (70.1%) completed the questionnaire and were included in subsequent analyses. The final participation rate was 91.1% in cases, 84.4% in exposed, 66.9% in controls.

The research was conducted in accordance with the Helsinki declaration and approved by the Catholic University Ethics Committee (ID3008, 16 April 2020).

The completed Strengthening the Reporting of Observational Studies in Epidemiology (STROBE) checklist is presented in Appendix A.

### 2.2. Questionnaire

In addition to socio-demographic data (gender, age, professional category, marital status, presence of children under 18 or elderly relatives who were not self-sufficient, presence of people who could be of help), the questionnaire included a list of 13 physical symptoms in the last 2 weeks that had been specifically designed for this investigation and comprised the symptoms most commonly associated with COVID-19 in the literature, and also a list of psychological symptoms. Goldberg Anxiety and Depression Scale (GADS) were used to evaluate the latter [23] (Italian version [24]). This questionnaire which consists of 9 binary-answered questions for each of the two subscales, was originally designed to be used by general practitioners for the screening of patients and is widely used in occupational medicine investigations on account of its simplicity, speed and reliability. The reliability of the sub-scales in this study was 0.762 for anxiety and 0.645 for depression. Subjects with an anxiety score of 5 points or more, or who score 2 points or more for depression have more than a 50% probability respectively of being diagnosed as anxious or depressed by a psychiatrist. This probability grows rapidly in proportion to the score [23].

Workers were also invited to express their perception of procedural justice (PJ) through the Colquitt questionnaire [25], Italian version [26], short form [27]. Procedural justice (three items, e.g., “Have you been able to express your views and feelings during those procedures?”) refers to processes and procedures employed to resolve conflicts or allocate resources [28]. The three items are answered according to a Likert scale of 5 points, ranging from 1 = “I strongly disagree” to 5 “I strongly agree”. The PJ score ranges from 3 to 15. In this study the reliability of the questionnaire measured by Cronbach’s alpha, was 0.855.

Occupational stress was measured using Siegrist’s effort/reward imbalance (ERI) model, which has been extensively used in HCWs [29,30,31]. The short form [32], Italian version [33], of the ERI questionnaire [34] is composed of 10 questions ranging on a 4-point Likert scale from “strongly disagree” to “strongly agree”. The effort subscale is based on three questions (e.g., “I have many interruptions and disturbances while performing my job”); the total score ranges from 3 to 12. The reward sub-scale is based on seven questions (e.g., “I receive the respect I deserve from my superior or a respective relevant person”); consequently, the score ranges from 7 to 28. The reliability of the two effort and reward sub-scales in this study was 0.783 and 0.843, respectively. Stress is measured as a weighted relationship between effort and reward. Conventionally a ratio value exceeding one is held to indicate a disparity between effort and rewards.

Sleep quality was measured using the short version of the Sleep Condition Indicator (SCI) [35] Italian version [36] with two questions (“how many nights a week do you have a problem with your sleep?”, and “to what extent has poor sleep troubled you in general?”) graduated on a 5-point Likert scale (from 4 to 0). SCI is a brief scale which measures sleep problems against the DSM-5 criteria for insomnia disorder. It has been used in large adult samples and occupational cohorts, including HCWs [37]. The sum of the answers gives a score ranging from 0 to 8, giving a general indication of the quality of sleep. In this study, Cronbach’s alpha for the questionnaire was 0.829. The whole questionnaire contained a total of 27 questions. The average time for completing the questionnaire was 5 min.

### 2.3. Statistics

The chi-square test was used to compare the characteristics of the three groups: positive HCWs, non-positive exposed workers and controls. By means of logistic regression, the odds ratio (OR) and the confidence interval (95%CI) of each symptom were assessed in the two groups that tested positive and negative, with the control group = 1.

Since self-reporting of symptoms is known to be influenced by psychosocial factors such as loneliness [38], occupational stress [39] or conditions of fatigue and sleep deprivation [40], we adjusted the multiple logistic regression calculation of the OR by introducing, in model I, age, gender, family status, the presence of children or older parents or relatives and loneliness or social isolation (expressed by the possibility of obtaining help in case of need). Model II analysis was adjusted to include also occupational stress (measured as effort/reward imbalance, ERI) and perceived procedural justice (PJ). Finally, Model III was adjusted to include sleep quality.

By analysing the variance (One-way Anova), we ascertained whether anxiety and depression were linked to one of the exposure groups (COVID-19 cases, exposed workers, controls). The Bonferroni test was then used to compare the three groups.

The association of socio-demographic factors (gender, age, family, children, parents/relations, people who could provide help), occupational variables (stress, organizational justice) and sleep problems with anxiety and depression was studied by linear regression. Using a stepwise method, we selected the variables significantly associated with mental health.

The effect modification by sleep quality (the potential effect modifier) on the relationship between ERI and PJ with anxiety and depression (both re-codified in binary terms, i.e., below/above the corresponding median values) was investigated by age and sex adjusted logistic regression analyses. In these analyses, data were expressed as odds ratio, 95% CI and *p* values. The odds ratios of having an anxiety and depression > median associated to 0.1 unit increase in ERI and 1.0 unit increase in PJ across predefined sleep quality values was investigated by the standard linear combination method. Data were analysed with the statistical package IBM/SPSS Statistics 23.0 (IBM, Armonk, NY, USA).

## 3. Results

The characteristics of the study participants are described in Table 1. The chi square test did not show significant differences in socio-demographic variables in the three groups (COVID-19 cases, exposed workers, controls). However, a significant difference was observed in the composition of the three groups as regards professional category and age group (Table 2). Infected workers were mainly nurses and young people.

As expected, a very significant difference for all the symptoms investigated was observed among the three groups (Table 3). All symptoms potentially attributable to COVID-19 were reported most frequently in workers whose test was positive. The commonest symptoms observed in infected workers were muscle pain (52.4%) and exhaustion (47,6%). Anosmia (42.7%) and dysgeusia (37.8%) were very frequent in positive HCWs, but rare in exposed and control HCWs. Fever was reported by 28% of infected HCWs, and diarrhoea by 24.4%. On average, COVID-19 cases had 3.8 ± 3.6 symptoms. Symptoms reported by test-positive HCWs are presented in a heat map that illustrates the symptoms reported concomitantly (Figure 1). 56.1% of the latter HCWs manifested fever and/or at least one respiratory symptom (sore throat, cough, expectoration, breathlessness), while the remaining 43.9% had no respiratory symptoms. Anosmia, exhaustion and muscle pain affected 62.2% of COVID-19 cases. Approximately one in three of the HCWs who tested positive (29.3%) never manifested any symptoms.

A considerable proportion of the HCWs who had unprotected exposure to COVID-19 patients (34.2%) presented symptoms. The most common complaints reported in this group were exhaustion, muscle pain, sore throat, cough and eye irritation. Physical symptoms were also present in 17.2% of the control HCWs who did not report direct unprotected contact with cases of the disease. The most common symptoms in the control group were the same as in the exposed/non-positive group.

A large number of the workers (about one in three) were affected by high occupational stress. In the three groups, no difference was observed in the prevalence of distressed workers (Table 3).

According to the criteria of the GADS questionnaire, 99 workers (16.6%) were affected by anxiety and 121 (20.3%) by depression. The prevalence of common mental disorders in the entire HCWs population is not higher than that recorded with the same GADS questionnaire during periodic medical visits to the workplace over the years [41] nor of what is generally observed in the workplace by the occupational doctor [42]. The share of anxious and depressed HCWs was greater in the test- positive group than in the other two groups (Table 3).

In positive workers, univariate logistic regression analysis revealed a very significant increase in the odds ratio of physical symptoms, anxiety, and depression (Table 4). In the same group of workers, the risk estimate was unchanged or even increased in multivariate models after adjustment for psychosocial factors (Model I), stress (Model II) and sleep problems (Model III) (Table 5). Compared to controls, the risk of developing anosmia for workers manifesting SARS_CoV-2 in the upper airways was 100 times higher, while that of developing dysgeusia was 50 times greater and the risk of developing fever, exhaustion and muscle pain was 20-fold (Table 5).

Workers who had been exposed, but were not positive at the RT-PCR test, were found to have an increased risk of exhaustion, muscle pain, sore throat, cough, and diarrhoea, as well as anxiety and depression, compared to unexposed controls (Table 4).

One-way Anova showed significant variance (*p* < 0.001) in the level of mental health in the three groups (COVID-19 cases, exposed workers, controls). Using Bonferroni’s test on multiple comparisons, a very significant difference between the controls and the other two groups was observed for anxiety and depression, while the mean scores did not differ significantly in COVID-19 cases and exposed workers.

Stepwise linear regression analysis identified the variables significantly associated with mental health (Table 6). Besides being present in the groups of HCWs who were test-positive or exposed to Covid-19, the level of anxiety was directly related to female gender, occupational effort and low sleep quality. It was also inversely related to procedural justice.

Similarly, the level of depression in workers depended on whether they were positive for the COVID-19 test or had been exposed to risk without adequate protection. It was positively associated with female gender, the effort made during work and the occurrence of sleep problems. It was also inversely related to reward and procedural justice.

In workers who tested positive, we studied the effect of sleep quality in the relationship between occupational stress and the level of anxiety. The effect modification by sleep score on the relationship between ERI and an anxiety score >median (dependent variable) was investigated in a multiple logistic regression model including the potential effect modifier (the sleep score), the risk factor (ERI) and their interaction term (sleep score x ERI) as well as age and gender. Thus, by the linear combination method, the age- and sex-adjusted odds ratio of an anxiety score >median associated to 0.1 unit increase in ERI was calculated at pre-defined values of the effect modifier (i.e., the sleep score). Finally, the odds ratios and the corresponding 95% confidence intervals calculated at predefined values of the sleep score were compared among them by calculating the *p* value for effect modification. The moderating effect of sleep on the relationship between work-related stress and anxiety is reported in Figure 2.

## 4. Discussion

This study provides valuable insights into COVID-19 symptoms in a large cohort of HCWs during the acute phase of the COVID-19 epidemic. It is unique in that an analysis was performed of both the physical and mental health conditions of an otherwise healthy population. The study demonstrated that the health of the HCWs deteriorated dramatically: only a small number had no physical or psychological symptoms. Numerous HCWs contracted SARS-CoV-2 infection while working in a hospital. Since the HCWs were predominantly of female gender, most of the infected workers were female.

Healthcare facilities are known to be at high risk due to infectious diseases and hospitals can play an important role in amplifying local outbreaks of SARS-CoV-2, thereby constituting a risk for elderly and vulnerable populations [43]. Early detection of COVID-19 in HCWs is therefore a particularly useful measure for patient and community health. An important result of our study was the detection of the type of symptoms present in active HCWs. Workers who had previously been healthy usually manifested only mild symptoms of the disease. Only 3 cases required hospitalization and all cases recovered. Consequently, the clinical picture differed from that of hospital patients.

The commonest COVID-19 related symptoms being reported in the literature are fever, cough or chest tightness and dyspnea [44]. Acute conjunctivitis, nasal congestion and sore throat are also frequently reported as sites of virus contact during the early stages of the disease [45,46]. Current COVID-19 HCWs screening guidance for RT-PCR [43] recommends making an assessment of fever and respiratory symptoms (cough, shortness of breath, or sore throat). In our cohort, screening only for fever and/or sore throat, cough, breathlessness or other respiratory symptoms might have failed to detect 43.9% of infected HCWs.

Some of the studies conducted on workers in accordance with indications from international bodies, have mainly evaluated respiratory symptoms. In a study on Chinese HCWs, the first three symptoms displayed before diagnosis of COVID-19 were fever (41.8%), lethargy (33.0%) and muscle pain (30.1%) [47]. Fever and cough were reported as early symptoms in mild COVID-19 cases [48], and fever occurred in the majority of individuals hospitalised for COVID-19 [49]. Telephone interviews of 48 HCWs residing in King County (WA, USA), with laboratory-confirmed SARS-CoV-2 infections showed that in this American occupational cohort the most common initial symptoms were cough (50.0%), fever (41.7%), and myalgia (35.4%) [50]. A Dutch study on early symptoms in HCWs initially covered only respiratory and general non-respiratory symptoms; the study was then adapted to include also anosmia, diarrhoea, nausea and extreme tiredness. In this cohort, anosmia was reported by 47% of test-positives and was strongly associated with SARS-CoV-2 positivity [51].

In our sample, fever occurred in approximately one in four cases and in some was only short-lived; only one in three HCWs manifested breathlessness. On the contrary, anosmia and dysgeusia were particularly strongly associated with test positivity. These symptoms appeared in almost half of our COVID-19 cases but were very rare in the other groups of workers. Univariate associations that were assessed by calculating the odds ratio revealed a marked increase of these symptoms in COVID-19 cases compared with controls. Even if the OR may slightly overestimate the risk evaluation, a very high increase in risk was observed by multivariate logistic regression. Adjustment for possible confounding factors resulted in an increase in the OR for these symptoms, which therefore appear to be pathognomonic of COVID-19. Recently, in fact, anosmia and ageusia emerged among the earliest symptoms of SARS-CoV-2 infection, possibly due to a disruption of the ciliary nasal epithelium mediated by the novel coronavirus [52,53]. A retrospective study on American HCWs who tested positive at RT-PCR confirmed that anosmia/dysgeusia are associated with a marked increase of odds of a positive test [54]. In the same study, HCWs reporting three or more symptoms had a significantly increased odds of having positive assays.

Tiredness and muscle pain, although significantly present in COVID-19 CASES, were sometimes reported also by workers from other groups. The smallest increase in the OR was for irritative eye symptoms in the COVID-19 cases group. This finding is similar to that reported by Tostman et al. in their study on Dutch HCWs [51], in which none of the respiratory symptoms were associated with SARS-CoV-2 positivity and sore throat was less common among test positives than controls.

Our findings can be used to improve public health guidelines on self-isolating individuals whose symptoms are indicative of COVID-19.

In our cohort, almost a third of COVID-19 cases (29%) were asymptomatic. It is well known that high viral loads may be detected soon after the onset of illness, also in persons who are non-symptomatic or only slightly symptomatic [55]. Interestingly, many HCWs who had symptoms and others who were only slightly symptomatic continued to work until the positivity of RT-PCR test was demonstrated, thus contributing to the spread of the disease in healthcare settings. This behaviour has been common in other HC companies. A recent study from the Netherlands showed that 63% of HCWs had continued to work despite mild symptoms [56]. A comprehensive screening of HCWs with minimal or no symptoms can be useful for protecting patients and hospital staff [57].

A significant number of exposed HCWs, and also controls were found to have some of the symptoms most commonly associated with COVID-19. These symptoms may be difficult for the occupational physician to interpret correctly. Some symptoms might have been caused by other respiratory infections; fatigue might have been due to the excessive workload of HCWs during the pandemic [58]; occupational stress could have contributed to determining psychosomatic symptoms [42]. However, these data underline the need to evaluate the mental health condition of HCWs during the pandemic. Moreover, taking into account that RT-PCR has a low sensitivity, especially during the early course of the disease [59], we cannot definitively rule out the possibility that some HCWs have escaped the correct diagnosis.

To the best of our knowledge, to date, no available study has simultaneously addressed the physical and mental health symptoms in HCWs at the height of the COVID-19 epidemic. In our study COVID-19-positive workers had a fourfold risk of being anxious and depressed compared to controls, while in exposed workers, the risk of anxiety and depression was doubled. An analysis of the factors influencing anxiety and depression showed that, in addition to exposure to the risk of COVID-19, occupational stress and the perception of poor procedural correctness were important. Assuming that exposure to risk is the causative factor, and neuropsychological problems are the consequence, sleep acted as a moderating factor between work-related stress and anxiety.

The anonymity of data collection in our study prevented us from examining the reported symptoms in relation to the previous health condition of the workers. However, the prevalence of anxiety and depression in test-positive workers was much higher than the values commonly recorded in the same population in previous years. We therefore conclude that testing positive for COVID-19 and being directly exposed to patients with COVID-19 without adequate personal protective equipment are important causes of alteration in mental health.

Several studies have indicated the presence of mental health problems in HCWs during the pandemic. Chinese studies observed a high level of arousal in HCWs [60] and estimated the prevalence of psychological abnormality at 14.1% [61], an anxiety rate ranging from 11% to 28%, and depression ranging from 43% to 46% [62,63]. One out of three HCWs suffered from insomnia [64]. A questionnaire survey in Singapore and India postulated that various symptoms of throat pain, cough and myalgia in HCWs may be over-represented as a result of the psychological stress [65]. Data on mental health of HCWs should be taken with caution, because all studies are cross-sectional and many have no control groups. Moreover, the majority of the available studies were conducted in Asia, limiting the current generalization of the results [66]. All of these studies are important for having drawn attention to the problem. HCWs should be given adequate social and mental health support [66,67], a necessary provision that is sometimes overlooked.

The causes of psychosocial disorders have rarely been studied. Some studies have associated the level of impairment with the geographical region and therefore the gravity of the epidemic [64], or with the type of job involving front-line or non-front-line workers [63]. Other factors associated with the severity of mental symptoms as mediator variables were selection of personnel, preventive interventions, resilience, and social support [68].

Previous occupational studies have shown that stress and sleep problems, particularly insomnia, are related to each other: stress causes poor sleep quality, and insomnia increases the perception of stress [69]. The metabolic effects of stress are mediated by sleep problems [69]. In this study, we hypothesized that exposure to risk of infection may induce psychological symptoms and, consequently, we assessed the moderating effect of sleep. Longitudinal subsequent studies could evaluate the existence of an inverse causality, that is to verify the hypothesis that anxiety may alter sleep and increase the perception of stress.

To date, no researcher has had the opportunity to study the association between contracting the disease and anxiety or depression, nor has any researcher tried to analyse the relationship between occupational stress, sleep problems and mental health in HCWs. Our study highlights the importance of offering psychological support especially to workers who tested positive for COVID-19 and/or to those who were exposed without adequate safety protection. It also stresses the importance of proper sleep hygiene since this can help limit worker anxiety. In our study, the levels of occupational stress and of anxiety and depression in workers not directly exposed to SARS-CoV-2 infection were not significantly higher than those commonly found in HCWs.

The main strength of this study comes from having studied the physical and mental health of HCWs at the height of the COVID-19 epidemic in the Latium Region. Although the study was limited to a single public healthcare company, there are no reasons to believe that the HCWs in this company are different from those of other healthcare companies. Since the workers who participated in this study had undergone health checks for many years with the use of questionnaires, they willingly accepted our request for information. Previous studies conducted in the same company show that the information provided through questionnaires is reliable and corresponds to the real situation.

## 5. Conclusions

Our findings indicate that the first symptoms in HCWs may not involve respiratory problems and that limiting screening to the latter might fail to detect almost half of the cases that tested positive for Covid-19. The presence of a significant number of asymptomatic infected HCWs must lead to the extension of screening to all workers who have had contact, even if they are asymptomatic. Public health measures such as the early detection of COVID-19 cases can be effective in containing a hospital-related outbreak. Furthermore, the mental health aspect is of the utmost importance, especially in HCWs who have been exposed to the virus or who have contracted the disease. Since sleep quality is an important anxiety moderator, health promotion intervention designed to reduce occupational stress levels and improve sleep quality is urgently needed. Longitudinal studies will help to understand the role that the pandemic has had on the mental health of HCWs, disentangling it from the numerous occupational stressors to which this category is exposed.

## Figures and Tables

**Figure 1 ijerph-17-05218-f001:**
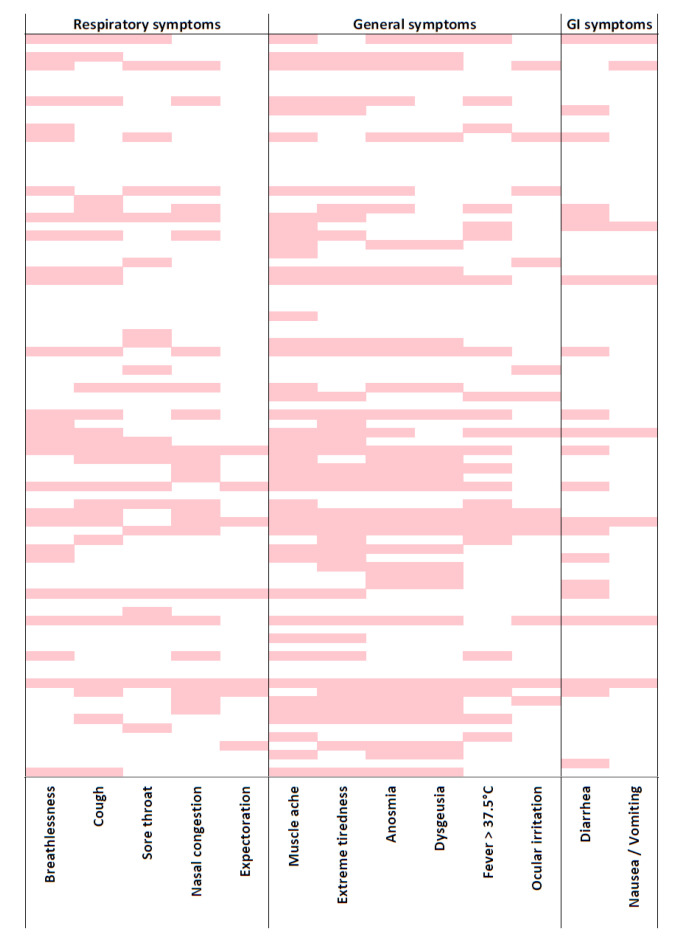
Heatmap of early symptoms reported by healthcare workers positive for SARS-CoV-2. Symptoms were grouped as respiratory, general, and gastrointestinal symptoms. Pink: symptom was present; white: symptom was absent. GI: gastro intestinal.

**Figure 2 ijerph-17-05218-f002:**
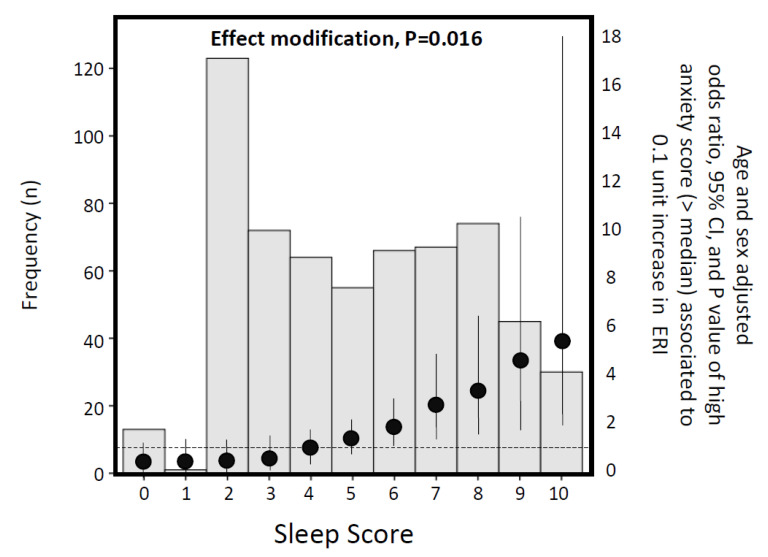
Histogram of the sleep score and age and sex adjusted increase of odds ratios (see black dots) and confidence intervals (95%CI) of high anxiety score (>median) associated with 0.1 increase in Siegrist’s effort/reward imbalance (effort reward imbalance) score across different values of sleep score.

**Table 1 ijerph-17-05218-t001:** Socio-demographic characteristics of the population.

Variable	Positive *n (*%)	Exposed *n (*%)	Control *n (*%)	Chi Square Test for Groups *p*
Groups	82 (13.8)	152 (25.5)	361 (60.7)	
Gender, female	56 (68.3)	104 (68.4)	257 (71.2)	n.s.
Family, single	64 (78.0)	116 (76.3)	273 (75.6)	n.s.
With Children	40 (48.8)	71 (46.7)	199 (55.1)	n.s.
With Parents/Relatives	62 (75.6)	131 (86.2)	294 (81.4)	n.s.
With Help	69 (84.1)	117 (77.0)	281 (77.8)	n.s.

n.s.: not significant.

**Table 2 ijerph-17-05218-t002:** Professional category and age class.

Variable	Positive *n (%)*	Exposed *n (%)*	Control *n (%)*	Chi Square Test for Groups *p*
Job category				<0.001
Physician	11 (13.4)	39 (25.7)	86 (23.8)	
Nurse	58 (70.7)	84 (55.3)	155 (42.9)	
Technician	7 (8.5)	14 (9.2)	43 (11.9)	
Clerk	2 (2.4)	7 (4.6)	47 (13.0)	
Other	4 (4.9)	8 (5.3)	30 (8.3)	
Age class (years)				0.012
<35	12 (14.6)	16 (10.5)	36 (10.0)	
36–45	17 (20.7)	40 (26.3)	61 (16.9)	
46–55	38 (46.3)	60 (39.5)	137 (38.0)	
>55	15 (18.3)	36 (23.7)	127 (35.2)	

**Table 3 ijerph-17-05218-t003:** Comparison of symptoms reported in the sub-groups of workers.

Symptom	Positive *n (%)*	Exposed *n (%)*	Control *n (%)*	Chi Square Test for Groups *p*
Fever	23 (28.0)	7 (4.6)	6 (1.7)	<0.001
Breathlessness	26 (31.7)	9 (5.9)	10 (2.8)	<0.001
Cough	26 (31.7)	16 (10.5)	15 (4.2)	<0.001
Expectorate	7 (8.5)	5 (3.3)	5 (1.4)	<0.002
Sore throat	20 (24.4)	20 (13.2)	14 (3.9)	<0.001
Anosmia	35 (42.7)	1 (0.7)	3 (0.8)	<0.001
Dysgeusia	31 (37.8)	5 (3.3)	4 (1.1)	<0.001
Nasal congestion	24 (29.3)	12 (7.9)	16 (4.4)	<0.001
Eye irritation	13 (15.9)	14 (9.2)	22 (6.1)	<0.013
Exhaustion	39 (47,6)	31 (20.4)	21 (5.8)	<0.001
Muscle pain	43 (52.4)	22 (14.5)	23 (6.4)	<0.001
Diarrhoea	20 (24.4)	13 (8.6)	15 (4.2)	<0.001
Nausea	7 (8.5)	3 (2.0)	5 (1.4)	<0.001
No symptom	24 (29.3)	100 (65.8)	299 (82.8)	<0.001
Distressed	24 (29.3)	47 (30.9)	109 (30.2)	n.s.
Anxious	29 (35.4)	29 (19.1)	41 (11.4)	<0.001
Depressed	30 (36.9)	40 (26.3)	51 (14.1)	<0.001

n.s.: not significant.

**Table 4 ijerph-17-05218-t004:** Univariate logistic regression analysis.

Symptom	Positive OR (95%CI)	Exposed OR (95%CI)	Control OR (95%CI)
Fever	23.065 (9.012–59.033) ***	2.856 (0.944- 8.645)	1
Breathlessness	16.296 (7.456–35.619) ***	2.209 (0.879–5.550)	1
Cough	10.710 (5.343–21.467) ***	2.714 (1.305–5.641) **	1
Expectorate	6.645 (2.054–21.504) **	2.422 (0.691–8.490)	1
Sore throat	7.995 (3.836–16.666) ***	3.755 (1.843–7.652) ***	1
Anosmia	88.865 (26.297–300.301) ***	0.790 (0.082–7.658)	1
Dysgeusia	54.250 (18.389–160.042) ***	3.036 (0.804–11.464)	1
Nasal congestion	8.922 (4.471–17.807) ***	1.848 (0.852–4.007)	1
Eye irritation	2.903 (1.395–6.042) **	1.563 (0.777–3.144)	1
Exhaustion	14.684 (7.914–27.246) ***	4.148 (2.296–7.494) ***	1
Muscle pain	16.203 (8.847–29.676) ***	2.487 (1.340–4.616) **	1
Diarrhoea	7.441 (3.615–15.317) ***	2.157 (1.001–4.652) *	1
Nausea	6.645 (2.054–21.504) **	1.434 (0.338–6.075)	1
No symptom	0.086 (0.050–0.149) ***	0.399 (0.259–0.615) ***	1
Distressed	0.957 (0.565–1.619)	1.035 (0.686–1.561)	1
Anxious	4.271 (2.446–7.457) ***	1.840 (1.095–3.092) *	1
Depressed	3.507 (2.047–6.007) ***	2.171 (1.361–3.463) **	1

Odds ratio (OR) and confidence interval (95%CI) of symptoms in the sub-groups of workers. *** *p* < 0.001; ** *p* < 0.01; * *p* < 0.05.

**Table 5 ijerph-17-05218-t005:** Multivariate logistic regression analysis.

Symptom	Model I OR (95%CI)	Model II OR (95%CI)	Model III OR (95%CI)
Fever	24.265 (9.296–63.338) ***	21.599 (8.215–56.793) ***	20.270 (7.688–53.442) ***
Breathlessness	17.503 (7.846–39.047) ***	15.746 (7.005–35.393) ***	14.931 (6.621–33.668) ***
Cough	10.974 (5.379–22. 386) ***	9.839 (4.767–20.307) ***	9.046 (4.359–18.773) ***
Expectorate	7.614 (2.178–26.615) ***	6.653 (1.881–23.526) **	6.364 (1.795–22.565) **
Sore throat	8.684 (4.064–18.557) ***	8.007 (3.717–17.248) ***	7.536 (3.485–16.296) ***
Anosmia	117.362 (31.583–436.122) ***	102.991 (27.676–383.267) ***	100.727 (26.520–382.578) ***
Dysgeusia	62.763 (20.247–194.557) ***	58.098 (18.580–181.665) ***	51.813 (16.573–161.980) ***
Nasal congestion	9.246 (4.549–18.792) ***	7.997 (3.879–16.485) ***	7.565 (3.651–15.675) ***
Eye irritation	3.884 (1.780–8.475) **	3.293 (1.488–7.289) **	3.087 (1.383–6.891) **
Exhaustion	18.121 (9.403–34.920) ***	16.414 (8.461–31.845) ***	16.111 (8.128–31.937) ***
Muscle pain	20.100 (10.498–38.486) ***	18.321 (9.487–35.383) ***	18.310 (9.221–36.360) ***
Diarrhoea	7.616 (3.641–15.933) ***	6.517 (3.080–13.788) ***	5.799 (2.718–12.374) ***
Nausea	6.267 (1.881–20.880) **	5.457 (1.611–18.491) **	5.129 (1.507–17.464) **
Anxious	4.395 (2.479–7.792) ***	3.715 (2.021–6.827) ***	3.704 (1.854–7.401) ***
Depressed	3.849 (2.208–6.707) ***	3.284 (1.811–5.957) ***	3.175 (1.596–6.315) ***

Odds ratio (OR) and confidence interval (95%CI) of symptoms in positive workers, adjusted for socio-demographic variables (model I), and additionally adjusted for job stress and organizational justice (model II) and sleep problems (model III). *** *p* < 0.001; ** *p* < 0.01; * *p* < 0.05. Model I adjusted for: gender, age, family, children, parents/relatives, persons who could help. Model II: also adjusted for job stress and procedural justice. Model III: also adjusted for sleep quality.

**Table 6 ijerph-17-05218-t006:** Association of socio-demographic characteristics, occupational variables (stress, organizational justice) and sleep problems with anxiety and depression.

Variable	Anxiety Standardized Beta	*p*	Depression Standardized Beta	*p*
Group *	−0.130	0.000	−0.138	0.000
Gender (female)	0.123	0.000	0.074	0.019
Age	-	-	-	-
Family	-	-	-	-
Children	-	-	-	-
Parents	-	-	-	-
Helping people	-	-	-	-
Effort	0.154	0.000	0.155	0.000
Reward	-	-	−0.111	0.002
Justice	−0.129	0.000	−0.070	0.047
Sleep quality	−0.531	0.000	−0.496	0.000
Coefficient of determination R^2^	0.458		0.430	

Linear regression, stepwise method. (*) 1 = positive; 2 = exposed; 3 = control.

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
