# Peer review of "Symptoms in Health Care Workers during the COVID-19 Epidemic. A Cross-Sectional Survey"

_ijerph, 2020, doi:10.3390/ijerph17145218_

Round 1

Reviewer 1 Report

Thankyou for the opportunity to review this manuscript. This is an interesting manuscript that is of great relevance to health care workers due to the current pandemic. The authors should be congratulated for undertaking such important research during this time, and the inclusion of a control sample was excellent for increasing the robustness of the results. Overall this was a good paper, however there are a number of details that could be added to aid clarity throughout the paper, and there were minor grammatical errors that should be addressed:

Abstract

  • Line 13 – It is unclear from the initial description of the sample whether the 82 health care workers have been diagnosed with COVID-19? And are the control sample also health care workers? Some greater detail here would be useful
  • Line 13 – a brief sentence or two describing the context of the research would add clarity to the abstract

Introduction

  • Line 34 – could some references please be provided of the countless studies that are mentioned as exploring COVID-19? This may be useful to readers who want to engage with other research similar to this paper
  • Line 45: In this paragraph, some greater discussion of the psychosocial factors that will be measured in this study would be beneficial, such as detail on why these symptoms might be greater in nurses anyway, and then exacerbated by COVID-19

Method

  • Line 85 – missing closing bracket to the list of socio-demographic data
  • Line 87 – should this be psychological symptoms or psychosocial symptoms rather than “psychic symptoms”
  • Line 96 – some greater detail on what is in the Colquitt questionnaire would be beneficial for readers who are not familiar with the topic of procedural justice. What do the questions ask/measure?
  • Similarly in line 101, more information on the actual questions in the ERI questionnaire would be beneficial. What do these questions specifically ask about occupational stress. Further, has this questionnaire been validated for health care workers?
  • For the sleep quality questionnaire (line 109) could information also be included on what exactly this questionnaire asks and if this has been validated or used in a population of health care workers.
  • Further, could some information be added to justify why the SCI was included to measure sleep quality, as this is a measure used to screen for insomnia rather than assess sleep quality. Some information to justify this would be useful in the methods section

Results

  • Table 3 - Where there any significant differences between the exposed and control groups for the symptoms? For example, exhaustion, sore throat, eye irritation, nasal congestion appear to have a difference between the groups
  • Figure caption for figure 1 would be beneficial to help orient the reader to the figure

Discussion

  • Line 224 - Given the prevalence of physical and mental health issues in health care workers, particularly shiftworkers, in general, can we say that this is an ‘otherwise healthy population’? Results show that 14% of the control sample were depressed, so it seems as though this sample is prone to low mental health and physical exhaustion etc

Author Response

Reviewer 1

COMMENT: Thankyou for the opportunity to review this manuscript. This is an interesting manuscript that is of great relevance to health care workers due to the current pandemic. The authors should be congratulated for undertaking such important research during this time, and the inclusion of a control sample was excellent for increasing the robustness of the results. Overall this was a good paper, however there are a number of details that could be added to aid clarity throughout the paper, and there were minor grammatical errors that should be addressed:

RESPONSE: We thank the reviewer for the attention with which he/she reviewed our work and for the interesting contributions he/she gave us, which can certainly help to improve the text and its usability by readers.

Abstract

Line 13 – It is unclear from the initial description of the sample whether the 82 health care workers have been diagnosed with COVID-19? And are the control sample also health care workers? Some greater detail here would be useful.

Line 13 – a brief sentence or two describing the context of the research would add clarity to the abstract

R.: We totally agree. We have tried to better explain the study in the abstract beginning, without adding too many sentences in order not to exceed the allowed length.

The text now is as it follows:

“In March-April 2020, the Covid-19 pandemic suddenly hit Italian healthcare facilities and in some of them many staff members became infected. 595 health care workers from a public company were tested for SARS-CoV-2 (82 positive) and asked to complete a questionnaire on early COVID-19 symptoms”

Introduction

Line 34 – could some references please be provided of the countless studies that are mentioned as exploring COVID-19? This may be useful to readers who want to engage with other research similar to this paper

R.: We are very pleased with this request because it has allowed us to insert some studies related to the subject matter, which came out after the submission of our work.

Line 45: In this paragraph, some greater discussion of the psychosocial factors that will be measured in this study would be beneficial, such as detail on why these symptoms might be greater in nurses anyway, and then exacerbated by COVID-19

R.: We agree. Work in healthcare activities exposes, as is well known, to significant biological and psychosocial risks. Studies have shown that traumatic events are associated with significant problems for nurses, doctors, or other professional categories. We have added two short sentences that refer to other studies on the subject

Method

Line 85 – missing closing bracket to the list of socio-demographic data

Line 87 – should this be psychological symptoms or psychosocial symptoms rather than “psychic symptoms”

R.: Thanks for reporting these typos.

Line 96 – some greater detail on what is in the Colquitt questionnaire would be beneficial for readers who are not familiar with the topic of procedural justice. What do the questions ask/measure?

R.: We gladly accepted this indication and gave a brief definition of procedural justice. We also reported a typical question.

Similarly in line 101, more information on the actual questions in the ERI questionnaire would be beneficial. What do these questions specifically ask about occupational stress. Further, has this questionnaire been validated for health care workers?

R.: We reported two typical questions of the relative questionnaire scales and cited some studies that used the questionnaire on HCWs

For the sleep quality questionnaire (line 109) could information also be included on what exactly this questionnaire asks and if this has been validated or used in a population of health care workers.

R.: SCI is a brief scale which measures sleep problems against the DSM-5 criteria for insomnia disorder. It has been used in large adult samples and occupational cohorts, including HCWs. We added this information in the manuscript.

Further, could some information be added to justify why the SCI was included to measure sleep quality, as this is a measure used to screen for insomnia rather than assess sleep quality. Some information to justify this would be useful in the methods section.

R.: Even though the questionnaire in its complete form was designed for the diagnosis of insomnia, the short form, with only two questions, refers exclusively to "sleep problems" and can therefore give an indication of the quality of sleep as a whole. We have synthetically reported this explanation.

Results

Table 3 - Where there any significant differences between the exposed and control groups for the symptoms? For example, exhaustion, sore throat, eye irritation, nasal congestion appear to have a difference between the groups.

R.: In fact, we observed that “A considerable proportion of the HCWs who had unprotected exposure to COVID-19 patients (34.2%) presented symptoms. The most common complaints reported in this group were exhaustion, muscle pain, sore throat, cough and eye irritation”. Moreover “Workers who had been exposed, but were not positive at the RT-PCR test, were found to have an increased risk of exhaustion, muscle pain, sore throat, cough, and diarrhoea, as well as anxiety and depression, compared to unexposed controls”. We concluded that “A significant number of EXPOSED HCWs, and also CONTROLS were found to have some of the symptoms most commonly associated with COVID-19. These symptoms may be difficult for the occupational physician to interpret correctly. Some symptoms might have been caused by other respiratory infections; fatigue might have been due to the excessive workload of HCWs during the pandemic [35]; occupational stress could have contributed to determining psychosomatic symptoms [21]. However, these data underline the need to evaluate the mental health condition of HCWs during the pandemic”.

Following the reviewer’s suggestion, we added this consideration: “Moreover, taking into account that RT-PCR has a low sensitivity, especially during the early course of the disease [Fang et al. 2020], we cannot definitively rule out the possibility that some HCW has escaped the correct diagnosis”

Figure caption for figure 1 would be beneficial to help orient the reader to the figure

R.: According to this suggestion, we modified the figure, placing the caption under the figure

Discussion

Line 224 - Given the prevalence of physical and mental health issues in health care workers, particularly shiftworkers, in general, can we say that this is an ‘otherwise healthy population’? Results show that 14% of the control sample were depressed, so it seems as though this sample is prone to low mental health and physical exhaustion etc.

R.: This observation is correct. As we indicated in the Introduction, HCWs are exposed to numerous occupational risks and often have suboptimal levels of physical and mental well-being. However, by the very fact of working, they are considered "healthy", even though we know that they are exposed to very serious phenomena such as distress, depression, burnout.

There is little doubt that HCWs are, on average, healthier than the general population. We are therefore convinced that it is appropriate to indicate them as "otherwise healthy" outside the pandemic and its physical and mental effects.

We believe that precisely because of being previously healthy, the physical symptoms of Covid-19 in these workers have been milder and qualitatively different than those described in hospital patients.

The psychological problems associated with the pandemic that many researchers have described are difficult to study with a cross-sectional design. So far there are no studies on the psychological effects of Covid-19 that have used a longitudinal design. We are convinced that this can be a useful development of our research.

Thanks to the reviewer's observation, we decided to end the paper with this sentence: “Longitudinal studies will help to understand the role that the pandemic has had on the mental health of HCWs, disentangling it from the numerous occupational stressors to which this category is exposed”.

Reviewer 2 Report

The author conducted an epidemiological study to show the psychological pressure of Italian medical staff. The subject is within the scope of this special issue. However, there is still much room for improvement in the study design and the statistical results. 

  1. Since this study wanted to show health care workers had higher psychological stress, why didn't the authors select non-health care workers or the public as the the reference group for comparison?
  2. Clarify is this a case-control study or cross-sectional study?
  3. If it is a case-control study, the reason to have two control groups seems meaningless and confusing. Suggest delete the so-call control group who were health care workers from the same Health Care Company and had not been exposed to virus. It is impossible for authors to confirm they really had not been exposed to virus in this pandemic.
  4. Table 1 and 2: Because the study used three groups, author only can use Chi-square test to show the distribution of frequency is different in the three groups. But the p-value did not provide valuable information, suggest delete.  
  5. To strengthen the reporting of this observational studies, author must use the STROBE guidelines were created to aid the author in ensuring high-quality presentation of the conducted observational study
  6. Delete figure 1 and 2. Prevent the use of fancy figures that do not provide sufficient useful information. 
  7. The odds ratio for many symptoms is very large and has very wide confidence interval. This may due to a lot of missing values in the questionnaire survey. The authors must provide the number of respondents for each symptom. 
  8. Too many typo.  

Author Response

Reviewer 2

The author conducted an epidemiological study to show the psychological pressure of Italian medical staff. The subject is within the scope of this special issue. However, there is still much room for improvement in the study design and the statistical results.

Since this study wanted to show health care workers had higher psychological stress, why didn't the authors select non-health care workers or the public as the the reference group for comparison?

Clarify is this a case-control study or cross-sectional study?

R.: The study was conducted by the doctor in charge of health surveillance of workers in a healthcare company during the explosion phase of the COVID-19 epidemic. The doctor sent an email to workers who tested positive for the RT-PCR test, inviting them to indicate recent symptoms. The email was also sent to a double number of workers who had had unprotected contact with Covid-19 patients and to a 6N number of other workers in the company. The design of the study is transversal, even if there are positive RT-PCR test workers, negative workers who fear that they have been infected because they have worked unprotected and workers who have worked in a standard way. The comparison of anxiety, depression, or sleep disturbance levels between HCWs and the general population, or that of stress levels and procedural justice in this or other types of work are not covered in this article.

If it is a case-control study, the reason to have two control groups seems meaningless and confusing. Suggest delete the so-call control group who were health care workers from the same Health Care Company and had not been exposed to virus. It is impossible for authors to confirm they really had not been exposed to virus in this pandemic.

R.: In a healthcare company that houses Covid-19 patients, during the pandemic all workers are professionally exposed to the risk. As is clearly explained in the article, most of the workers were exposed in standard protection ways, but a minority had unprotected exposure. Of these workers, some had a positive RT-PCR test. In the cross-sectional study, the symptoms of the three groups of workers were compared.

Table 1 and 2: Because the study used three groups, author only can use Chi-square test to show the distribution of frequency is different in the three groups. But the p-value did not provide valuable information, suggest delete.

R.: The chi square test has been used to test for homogeneity of binary variables. We did not observe significant differences in socio-demographic variables in the three groups (COVID-19 CASES, EXPOSED WORKERS AND CONTROLS, Table 1) and this means that the three groups are homogeneous. However, a significant difference was observed in the composition of the three groups as regards professional category and age group (Table 2). Infected workers were mainly young nurses. This result, which is clearly expressed in the tables and reported in the text, is not without meaning.

To strengthen the reporting of this observational studies, author must use the STROBE guidelines were created to aid the author in ensuring high-quality presentation of the conducted observational study

R.: We thank the reviewer for their kind indication. We had followed the STROBE checklist. We have attached this checklist as a supplement to the paper.

Delete figure 1 and 2. Prevent the use of fancy figures that do not provide sufficient useful information.

Response: We totally agree that the figures must be justified. In the previous version of the article we have not correctly explained the meaning of the figures. In this version, we have explained how figure 1 allows, through a heat map, to immediately see the association of symptoms which is often focused more on general symptoms than on respiratory ones.

Similarly, the Figure 2 illustrates the effect modification by sleep score on the relationship between ERI and an anxiety score >median (dependent variable), which was investigated in a multiple logistic regression model including the potential effect modifier (the sleep score), the risk factor (ERI) and their interaction term (sleep score x ERI) as well as age and gender. Thus, by the linear combination method, the age- and sex-adjusted odds ratio of an anxiety score >median associated to 0.1 unit increase in ERI was calculated at pre-defined values of the effect modifier (i.e. the sleep score). Finally, the odds ratios and the corresponding 95% confidence intervals (see black dots) calculated at predefined values of the sleep score were compared among them by calculating the P value for effect modification. By means of the figure, this very complex information is summarized and made immediately available to the reader.

The odds ratio for many symptoms is very large and has very wide confidence interval. This may due to a lot of missing values in the questionnaire survey. The authors must provide the number of respondents for each symptom.

R.: The study was conducted exclusively on workers who answered all the questions. The system does not allow participant to skip a response, or to stop the test and resume it later. Consequently, workers who did not complete the survey were discarded, as explained in the methods section. There were no missing data.

Too many typo.

R.: We hope we have corrected them all. 

Reviewer 3 Report

Thank you for the opportunity to review your work report the experience of treating healthcare workers in occupational health during the early stages of the COVID-19 epidemic. I have only minor comments and these are detailed in the attached file annotating the work for your convenience.

Author Response

Reviewer 3

(changes were directly indicated in the manuscript).

Responses: We thank for the attention with which the work has been reviewed.

We gladly accepted the invitation to specify better in the text, at Line 70/71, that the workers of the "exposed" group were "occupationally exposed whilst unprotected".

The reporting period for symptoms (l.87) was 2 weeks.

We also corrected (l. 159) the malposition of the "fever" symptom and we checked the indications shown in figure 1 and table 3. The correct translation of the Italian term we used is “breathlessness”. We also reported the legend in figure 1, which for a typo was previously absent.

With reference to Line 294 and 341, the reviewer correctly underlines the possibility of reverse causality. We fully agree on the need to better explain this point and to clarify that a cross-sectional study cannot infer causality.

Thanks again to the reviewer who reported us some typos. We found other errors in the text and corrected them.

Round 2

Reviewer 2 Report

1. Correct CI95% to 95% CI. 2. An OR of 100.7 is quite usual in epidemiological studies that might be caused by improper selection of control group. If possible, author could calculate the relative risk (RR) of prevalence using the modified Poisson regression with robust error variance. 3. Authors did not reply how can they confirm the controls did not have COVID-19? Since the controls did not receive tests for COVID-19 that will introduce selection bias. 4. The supplementary file of STROBE Statement is too rough. If authors did calculate the study size, provide the f, alpha, beta, p1, and p2. 5. Table 5, delete "NO SYMPTOM"

Author Response

Comments and Suggestions for Authors

We are very pleased with the reviewer's timely observations which have allowed us to clarify some points that had not been well expressed in previous versions of the manuscript.

Taking advantage of these indications, we have changed the description of the case series. The cross-sectional study incorporates a comparison of cases and controls.

In the STROBE sheet we therefore followed the recommended epidemiological criteria for both types of studies

  1. Correct CI95% to 95% CI.

Response: Thanks to this suggestion, we have corrected the text.

  1. An OR of 100.7 is quite usual in epidemiological studies that might be caused by improper selection of control group. If possible, author could calculate the relative risk (RR) of prevalence using the modified Poisson regression with robust error variance.

Response: The referee is absolutely right when saying that an OR of 107.7 is rather high. To comply with the request of the reviewer we specifically checked whether the key variables follow a Poisson distribution (by the Kolmogorov Smirnov test) and unfortunately this is not the case. Thus, we cannot perform the interesting analysis indicated by the referee.

  1. Authors did not reply how can they confirm the controls did not have COVID-19? Since the controls did not receive tests for COVID-19 that will introduce selection bias.

Response: We apologize with the referee whether this point is still unclear. The study adopted a cross-sectional design to know the health status of a local health company HCWs. The focus of the investigation were workers who had unprotected contacts with proven Covid-19 cases. All workers who tested positive for the RT-PCR test were contacted. Their health was compared to that of workers who had unprotected exposure, but were negative on the test, and workers who had no unprotected exposure, who were also negative on the test. The workers contacted (810 people) represented 66.3% of the population undergoing health surveillance for occupational risks in the company. Although selection bias could be never definitively ruled out, controls were comparable, according to the basic principles indicated by Wacholder et al. (1992): study base, as all comparisons were made within the study base; confounding, as the comparisons were not distorted by other factors; accuracy, as all HCWs performed the same test. According to your suggestion, we have changed the study description.

  1. The supplementary file of STROBE Statement is too rough. If authors did calculate the study size, provide the f, alpha, beta, p1, and p2. 5. Table 5, delete "NO SYMPTOM"

Response: I We indicated in the checklist that to evaluate the sample size we applied the formula suggested by Pocock:

N= f (α/2, β) * [p1 * (100- p1) + p2 * (100- p2)] / (p2 – p1)2

If we calculate the probability of finding a symptom in the CASE group and in the CONTROL group, we can calculate the size of the population, placing a significance level (alpha) at 5% and a power (1-beta) at 90%.

For a symptom such as anosmia, which has a prevalence of 42% in cases and 0.8% in controls, the minimum sample size involves 16 cases and as many controls, total = 32 observations.

For a symptom such as anxiety, which has a prevalence of 35% in CASES and 11% in CHECKS, the required dimensions are 60 per group, total 120 observations.

All calculations were carried out with the help of the automatic calculator: Sealed Envelope Ltd. 2012. Power calculator for binary outcome superiority trial. Available online at: https://www.sealedenvelope.com/power/binary-superiority/ [Access May 26, 2020].

We included these details in the STROBE checklist

We deleted NO SYMPTOM in Table 5.